# The Beliefs and Attitudes of Cypriot Physical Therapists Regarding the Use of Deep Friction Massage

**DOI:** 10.3390/medicina55080472

**Published:** 2019-08-12

**Authors:** Alexios Pitsillides, Dimitrios Stasinopoulos

**Affiliations:** 1Department of Health Sciences, School of Sciences, European University Cyprus, Nicosia 1516, Cyprus; 2Cyprus Musculoskeletal and Sports Trauma Research Centre (CYMUSTREC), Nicosia 1516, Cyprus

**Keywords:** Cyriax method, transverse frictions, deep friction massage, deep transverse massage, friction massage, chronic pain, chronic musculoskeletal pain

## Abstract

*Background*: Deep friction massage (DFM) is a widely used technique by physical therapists worldwide for chronic pain management. According to Dr. James Cyriax, compliance with the proposed guidelines is vital to obtain the desired therapeutic results. *Objectives*: This study explored the beliefs and attitudes of Cypriot physical therapists to DFM and their compliance with the suggested guidelines to identify any empirical-based application patterns and compare them to the suggestions of Cyriax. In addition, the prevalence of DFM use in clinical practice in Cyprus was investigated. *Methods*: Questionnaires, consisting of 18 multiple choice questions and a table of six sub-questions, were distributed to 90 local physical therapists. *Results*: A total of 70% of respondents declared that they perform DFM in their daily practice. The respondents answered 11 out of the 19 technical questions in compliance with the guidelines. *Conclusion*: The study revealed the DFM application pattern of Cypriot physical therapists. The compliance percentage of this pattern to Cyriax guidelines was 58% in general and 62.5% for patients with chronic conditions.

## 1. Introduction

Physical therapists worldwide treat chronic pain on a daily basis. A widely known technique for treating chronic musculoskeletal pain is deep friction massage (DFM) [1]. Although Mennell proposed a special massage technique called “frictions” in 1982 [2], DFM was officially introduced to and popularized in the clinical world by Dr. James Cyriax in 1984. According to Cyriax, DFM can be distinguished from general massage as it reaches the deep structures of the body, such as the ligaments, tendons, and muscles [1]. DFM has four key aims: (1) To induce pain relief, (2) to produce therapeutic movement, (3) to produce traumatic hyperemia in chronic lesions, and (4) to improve function [3,4,5]. The technique can be found in the literature under several names. However, incorrect terminology may negatively affect the results of the technique through differences in execution [1,6]. In the authors’ opinion, the general term should be transverse friction massage (TFM) with individual terms used according to the chronicity of the injury. Specifically, the terms gently transverse massage (GTM) for acute injuries and deep transverse friction (DTF) for chronic injuries are apt descriptions of the procedure that must be followed in each case [7].

However, the effectiveness of DFM has not been clearly documented [4,5]. A considerable number of reviews concluded that sample sizes and methodological limitations were the main limiting factors. One such limitation is the lack of standardization of the DFM protocol [3,4,5,6].

In 2017, Chaves et al. conducted a cross-sectional study of 478 respondents to determine the prevalence of DFM in clinical practice in Portugal. Further, they aimed to characterize the application parameters used by local physical therapists and identify empirical model-based patterns of DFM application in degenerative tendinopathy. The study concluded that Portuguese physical therapist application patterns were in conflict with the rules of Cyriax. As alluded to above, the aim of their study [8] was not to identify the effectiveness of DFM, but to measure their subjects’ compliance to Cyriax guidelines, as according to the proposed guidelines, absolute compliance is vital for successful therapeutic results.

Taking the above-mentioned study into consideration, we designed and conducted the present study on the Cypriot therapeutic community. The purpose of the study was to explore the beliefs and attitudes of Cypriot physical therapists on DFM. The results will be beneficial on a local level, as they could help to highlight whether there is a need to improve application patterns locally. Furthermore, helpful conclusions may be deducted through comparisons with similar studies conducted in other countries. As an example, non-compliance to Cyriax’s protocol in clinical practice may translate into a need for revised, more comprehensive guidelines or into a need for improvement and modernization of the current guidelines [7]. Furthermore, the results of the study will add to the existing body of evidence on DFM. The present study is innovative as no other work in the literature has explored this particular issue in Cyprus.

## 2. Materials and Methods

### 2.1. Aims of the Study

The aims of this study were firstly, to determine the prevalence of DFM used in clinical practice in Cyprus, secondly, to explore the application of parameters used by local physical therapists, and more specifically, to check if the guidelines proposed by Cyriax were being applied by Cypriot physical therapists, and finally, to identify any empirical model-based patterns.

### 2.2. Questionnaire Design

The questionnaire was designed as part of a postgraduate degree research project based on the relevant literature. Following corrections from the student’s advisor, the questionnaire was finalized. It was then distributed to five physical therapists to confirm that the questions were easily comprehensible; subsequently, some amendments were made. The questionnaire used closed-ended questions with an open comments section at the end of each question.

### 2.3. Questionnaire Sharing

The questionnaire was distributed to 90 of the 943 registered Cypriot physical therapists. The process started on 28 November 2018 and ended on 28 January 2019. Some of the questionnaires were completed by local physical therapists who were present at the annual meeting of the Cyprus Physical Therapists’ Association. The rest were distributed by the author to private physical therapy clinics.

In order to be considered eligible to take part in the study, one had to be a registered physical therapist. Physical therapy students or physical therapy assistants were not eligible to participate. Once the questionnaire was distributed, no further verbal communication between the participants and the author was allowed. The questionnaire was completed in the presence of the author to minimize the likelihood of the results being altered by copying or searching for the correct answer using the internet.

### 2.4. Ethics

All questionnaires were completed anonymously. Precautions were taken to ensure that all participating physical therapists were fully informed of the purpose and parameters of the questionnaire and consented to the completion of the questionnaire. Relevant information concerning the exact purpose of the study, information on the authors, and the fact that the results might be published were provided on the first page of the questionnaire. Any participant could decline to take part in the survey after consenting, if they so wished.

### 2.5. Questionnaire Content

The first page of the questionnaire described the purpose and subject of the survey. It further clarified the anonymity of the answers. The first column of Table 1 below lists the 18 multiple-choice questions and a table of 6 sub-questions related to Question 7 contained in the questionnaire.

### 2.6. Data Management and Analysis

The data were tallied by simple counting of the responses for every possible answer in each question. Due to the fact that the respondents were allowed to choose more than one answer or even skip a question, the total number of given answers was also counted for every question individually. The percentage of every answer was calculated in relation to the total number of collected answers to that specific question. The number of each answer was multiplied by one hundred and then divided by the sample size of that question.

## 3. Results

Table 1 summarizes the participants’ professional characteristics. Table 2 displays the answers to all the questions answered by respondents. Table 3 consists of three columns, which display (1) the questions, (2) the most frequent answer, and (3) a comparison of this study’s results with the guidelines proposed by Cyriax (shown in the following column). The majority of participants were graduate physical therapists (BSc) with more than 10 years of experience who claimed to specialize in musculoskeletal rehabilitation. Most participants (63 out of 90, 70%) also declared that they used DFM in their daily practice. A total of 59% (51 out of 81) of the sample were taught the technique in their pre-graduate education.

## 4. Discussion

### 4.1. General Discussion

Out of the 90 questionnaires distributed, 27 were not completed beyond Question 4. This most likely suggests that 30% of the sample did not perform DFM in their daily practice or they did not specialize in musculoskeletal injuries. However, is not possible to say that all of those who did not proceed beyond Question 4 did not use the technique, as they may have chosen to stop for other reasons, such as lack of motivation. We believe that asking our subjects if they wished to participate before we shared the questionnaire minimized this possibility.

Seventy percent of our sample declared that they performed the technique and specialized in musculoskeletal injuries. The remaining 63 questionnaires were analyzed in order to investigate the beliefs of Cypriot physical therapists on DFM.

The majority of respondents were graduate physical therapists who mainly specialized in musculoskeletal treatments and applied DFM in their daily practice. Respondents were mainly taught the technique during their university studies.

One of the aims of the study was to identify the presence of any empirical model-based patterns. The questionnaire results revealed that DFM was applied by local physical therapists in a different way to the technique proposed by Cyriax. This empirical model-based pattern is described in the “Respondents” column in Table 3.

The local application pattern differed as compared with the guidelines for seven parameters. Firstly, local physical therapists combined DFM with a variety of other techniques, not only manual therapy, as proposed. Secondly, Cypriot physical therapists applied the technique every 48 h in cases of acute injury, rather than daily. In addition, a stricter interval (every 48 h) than that suggested by Cyriax (48 h minimum) was preferred for chronic conditions. With regard to the speed of application, Cypriot physical therapists claimed to use a slow but gradually increasing speed, whereas Cyriax suggests a slow speed and clarifies that a faster speed mostly affects the superficial tissues [1,3]. When a muscle is the target, local physical therapists preferred to apply DFM with the tissue in an accessible rather than a shortened position. Attention must be given to the therapist’s hand position to avoid fatigue. Out of the six hand positions proposed by Cyriax, only two were used by Cypriot physical therapists. The last difference identified concerned the intermediate material used. The respondents stated that they preferred to use an anti-inflammatory gel instead of not using anything, as suggested by Cyriax.

Apart from Questions 7a–c, the survey could be analyzed as an independent questionnaire for chronic pain treatment. In such a case, sixteen technical questions concerned the beliefs and attitudes of local physical therapists regarding chronic pain management. The results of a questionnaire in this scenario would lead us to the same conclusions as the original questionnaire used in the study. In more detail, 10 out of the 16 questions would be in compliance with the suggested guidelines. Even if chronic pain management was investigated in isolation, a substantial percentage of physical therapists in Cyprus did not comply with the suggested guidelines.

Similar to a study published [8] in Portugal, our results demonstrate that the daily application of DFM in Cyprus is not in complete compliance with the proposed instructions of Cyriax. This could be interpreted in different ways. Firstly, it could be that the guidelines need to be revised and improved. The second reason could be that better instructions are needed with regard to this particular technique. Alternatively, the technique proposed by Cyriax may not provide clinicians with adequate therapeutic results, leading to modifications to the technique during daily practice.

The observed non-compliance with the instructions suggested by Cyriax is not only observed among Cypriot physical therapists. As mentioned previously, Portuguese physical therapists were also found to apply DFM in their own ways [8]. In addition, a lack of awareness or differentiation between some technical details of DFM can be observed in the literature. For example, in chronic conditions, the technique must be applied for 10 min after analgesia [1]. However, this proposed duration was neither used by the subjects of this study nor by the authors of previous studies related to DFM [6,9,10,11,12,13,14,15,16,17,18].

To date, the effectiveness of this technique has not been documented sufficiently [4,5]. Combined with the observed tendencies of clinicians from different countries, the question of whether the suggested guidelines need to be modified or whether this technique should be used at all has been raised. It is the authors’ opinion that DFM should undergo important modifications and a thorough investigation on its effectiveness should be made before it can be recommended to the clinical world [7].

DFM is an old technique, thus, its characteristics are strongly challenged by the better understanding of certain pathologies [7]. For example, the currently suggested technique for tendinopathy is considerably different from that suggested by Cyriax. The use of DFM in such a condition would only result in the breakage of any adhesions, thereby producing traumatic hyperemia. This approach, in cases of tendinopathy, does not target the ability of tendons to store and release energy [19] or their neuroplasticity [20], both of which are vital in tendon rehabilitation.

Our study adds to the body of evidence on DFM. It is the second study to reveal non-compliance of physical therapists to the guidelines. Further, in addition to the observed lack of awareness or the differentiation of some technical details of DFM in the literature, we conclude that physical therapy professionals may have to move on from this traditional technique.

Bearing in mind the insufficiently documented effectiveness of DFM and having observed that physical therapists in different countries tend to change the proposed guidelines—possibly in an effort to improve their therapeutic results—the authors conclude that DFM should, at present, not be included as a first-line treatment.

### 4.2. Study Limitations

This study is not free of limitations. It concerns the beliefs of Cypriot physical therapists and, therefore, the results cannot be generalized to other countries. Another limitation is that there was no randomization. A third limitation might be the fact that respondents were allowed to choose more than one answer in some questions or even skip one. Despite the use of an equation to calculate the percentage of each answer in relation with the sample size of each question individually, this might have introduced bias associated with double-counting responses.

## 5. Conclusions

In the current study, 58% of respondents were found to apply DFM in compliance with the suggestions of Cyriax. However, when treating chronic pain, 62.5% complied with the suggested guidelines.

The study revealed the application pattern of DFM followed by Cypriot physical therapists. This pattern differs from the suggestions of Cyriax for seven of the measured parameters.

## Figures and Tables

**Table 1 medicina-55-00472-t001:** Professional characteristics of the sample (*n* = 90). DFM: Deep friction massage.

Question	Final Sample	
Academic qualifications	*n*	%
BSc	62	69
Master	26	29
PhD	2	2
Other	0	0
Total	90	100
**Years of experience**		
1–3	16	18
3–5	16	18
5–10	21	23
Other	37	41
Total	90	100
**Field of specification**		
Cardiorespiratory	9	7.5
Musculoskeletal	74	61.6
Neurological	25	20.9
Other	12	10
Total	120	100
**Use of DFM in daily practice**		
Yes	63	70
No	27	30
Total	90	100
**Learned the technique from:**		
Book	14	16
University	51	59
Seminar	17	20
Other	4	5
Total	86	100

**Table 2 medicina-55-00472-t002:** Questions and answers given by participants (*n* = 90).

Question	Final Sample	
	*n*	%
**6. Do you use the technique as…?**		
A single treatment	3	5
In combination with another technique (which one)	43	52
In combination with physical modalities	33	40
Other	2	3
Total	81	100
**7a. Acute phase (friction intensity)**		
Gentle	48	77.4
Deep	2	3.3
Other	12	19.3
Total	62	100
**7b. Acute phase (frequency)**		
Daily	8	11.4
Every 48 h	33	47
Once per week	9	12.8
Other	9	12.8
Total	69	100
**7c. Acute phase (duration)**		
Until analgesia plus 10 maneuvers more	23	38.2
10 min after analgesia	17	28.8
Other	19	32
Total	59	100
**7d. Chronic phase (friction intensity)**		
Gentle	3	4.8
Deep	57	90.4
Other	3	4.8
Total	63	100
**7e. Chronic phase (frequency)**		
Daily	15	23.4
Every 48 h	36	56.3
Once per week	9	14
Other	4	6.3
Total	64	100
**7f. Chronic phase (duration)**		
Until analgesia plus 10 maneuvers more	27	43.5
10 min after analgesia	26	42
Other	9	14.5
Total	62	100
**8. Where do you apply DFM?**		
Around the point of pain	27	32
On the exact point of pain	54	64
Other	3	4
Total	84	100
**9. What is the direction of the applied force?**		
Transverse	51	74
Parallel	12	17
Other	6	9
Total	69	100
**10. What is the ideal depth?**		
Enough to ensure the compression and friction of the target tissue	40	60
Until pain reaches 3/10 on a Visual Analog Scale (VAS)	21	31
Other	6	9
Total	67	100
**11. What criteria do you use to choose the applied depth?**		
Injury phase	35	37
Pain scale	39	41
Experience	19	20
Other	2	2
Total	92	100
**12. Speed of application**		
Slow, gradually increase	46	72
Fast	8	12
Other	10	16
Total	64	100
**13. Patient positioning**		
Comfortable position	46	67
With the area of application stretched	18	26
Other	5	7
Total	69	100
**14. If the target tissue is a muscle, it should be positioned**		
Stretched	27	37
Accessible	39	53
Shortened	4	5
Other	3	5
Total	73	100
**15. If the target tissue is a ligament, it should be positioned**		
Stretched	24	35
Accessible	37	54
Shortened	7	10
Other	1	1
Total	69	100
**16. If the target tissue is a tendon, it should be positioned**		
Stretched	27	38
Accessible	35	49
Shortened	7	10
Other	2	3
Total	71	100
**17. If the target tissue is a tendon with sheath, it should be positioned**		
Stretched	31	49
Accessible	26	41
Shortened	5	8
Other	1	2
Total	63	100
**18. What grip do you use?**		
With the edge of the last phalanx and the last interphalangeal joint bent	24	28.5
With the edge of the last phalanx and the last interphalangeal joint in flexion	31	37
With tools	24	28.5
Other	5	6
Total	84	100
**19. Intermediate material**		
Oil	24	28
Gel	10	12
Anti-inflammatory cream	22	25
Other	30	35
Total	86	100

**Table 3 medicina-55-00472-t003:** Questions, answers, and guidelines.

Question	Respondents	*n*	%	Cyriax
6. Application of DFM in combination with any other technique/single therapy/physical modalities.	In combination with another technique	43	52	Combined with manual therapy
7a. Acute injury, friction intensity	Gently	48	77.4	Gently
7b. Acute injury, frequency	Every 48 h	33	47	Daily
7c. Acute injury, friction duration	Until analgesia plus 10 maneuvers more	23	38.2	Until analgesia plus 10 maneuvers more
7d. Chronic injury, friction intensity	Deep	57	90.4	Deep
7e. Chronic injury, frequency	Every 48 h	36	56.3	Every 48 h minimum
7f. Chronic injury, friction duration	Until analgesia plus 10 maneuvers more	27	43.5	Until analgesia plus 10 min more
8. Spot of application	Exact spot of pain	54	64	Exact spot of pain
9. Applied force direction	Transverse	51	74	Transverse
10. Ideal depth of friction	That ensures tissue compression	40	60	That ensures tissue compression
11. Criteria for ideal depth	Injury chronicity and patient pain.	39	41	Injury chronicity and patient pain.
12. Application speed	Slow, gradually increasing	46	72	Slow
13. Patient position	Comfortable	46	67	Comfortable
14. Muscle position while applying DFM	Accessible	39	53	Shortened
15. Ligament position while applying DFM	Accessible	37	54	Accessible
16. Tendon position while applying DFM	Accessible	35	49	Accessible
17. Tendon with sheath position while applying DFM	Stretched	31	49	Stretched
18. Preferred hand grip	2 out of 6 suggested by Cyriax	31	37	Cyriax suggested 6 hand grips
19. Intermediate material	Anti-inflammatory gel	22	25	No intermediate material

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
