# Peer review of "The Beliefs and Attitudes of Cypriot Physical Therapists Regarding the Use of Deep Friction Massage"

_medicina, 2019, doi:10.3390/medicina55080472_

Round 1
Reviewer 1 Report
I thought the topic of research was interesting with implications on physiotherapists’ clinical practice in relation to the use of Deep Friction Massage in Cyprus. However, the writing of the article could be improved (English language) for clarity (for e.g., specify “local” in line 17; line 21 unclear “as regards Cypriot”; line 226: there’s a reference to chapter, which is unclear), spelling / typographical errors (for e.g., line 188 “Cyprus” instead of “Cyrpus”) and grammar (for e.g., line 11 in Abstract – the author probably wants to write “by” instead of “to” the physical therapist; line 30 “physical therapists treat patients with chronic pain on a daily basis” instead of “physical therapists worldwide face chronic pain in a daily basis”). The use of figures (21 pie charts in total) could be reduced and the figure legends are not very clear. It would be useful to use figures to report results only where necessary – other findings could be reported in words. The discussion could be more focussed with references to evidence for the clinical effectiveness of Cyriax. Line 217-218: this doesn’t appear to be a limitation considering the aim of the study. The conclusion could be concise and punchy in line with the aim of the study.
Author Response
Firstly, we would like to thank you for spending time to read and comment to our paper. Your suggestions have been very helpful and had been taken in serious concern. In our try to improve our paper we have corrected all the language errors you have suggested and few more. Concerning your comment about the large number of figured we decided to illustrate only the questions where our sample’s answer was in conflict with the guidelines. We also changed the format from pie chart to columns in an effort to present our results in a better way. About your comment regarding our discussion and the fact that it could be more focused to the effectiveness of Cyriax technique, we would like to point out that this could not be done since there is a gap in the literature about the effectiveness of the technique.
Reviewer 2 Report
Generally, I think this is a useful topic on which to conduct research and there is a definite lack of evaluation of technique and effectiveness of this technique. However, there are lots of issues with the way the paper was written and the way the analysis was presented and discussed.
Firstly, the paper needs more thorough proof-reading as some sentences were lost in translation and difficult to follow. Also there are some grammatical changes needed, which I have tried to point out but might have missed a few.
Abstract:
This is a little difficult to follow for the reader and there are some grammatical and sentence construction errors as follows:
Lines 12/13 - should read '.... complicance with the proposed....' Also same at the end of line 14.
Line 14 - you can't start a sentence with 'Identify'. It needs to say 'We identify'.
Line 16 - should read 'Finally we determine....'
Line 22 - 'with Cyriax guidelines'
Introduction:
At the start no mention is made of the Cochrane Review and there is perhaps more research to include.
https://doi.org/10.1002/14651858.CD003528.pub2
Line 31 - define DFM in full again for the main body of the text.
Lines 35/36 - reword this sentence so that its meaning is clearer.
Line 53 - 'compliance with'
Line 55 - Again, the sentence needs re-wording. Previously mentioned what? In fact, you've only actually described in detail one study.
Line 58 - You have interchanged the terms Physiotherapist and Physical Therapist. You need to stick to one term.
Line 60 - Surely the second benefit would be that this study would add to the body of evidence?
Line 61 - comparing
Line 62 - suggested rewording '...might highlight the need for revised guidelines...'
Line 63 - Need for
Line 64 - I would word it by saying this study is novel in that no other similar work has been carried out in Cyprus.
Methods:
At the start of this you talk about the aim, but in fact you have three aims.
Again, you mis up the terms for Physiotherapists.
Line 74 - 'As part of...'
Lines 77/78 - Did you mean to say that an open free-text comments section was included?
Line 80 - How did you select the sample? How was the randomisation done (e.g. by computer)? What sort of randomisation was it? Stratified? Block?
Line 92/3 This sentence is not easy to understand and needs rewording.
Line 94 - 'were completed'. Also, remove 'All the' from the next sentence.
Line 96 - 'consented to the completion of the questionnaire'
Line 97 - take out the capital letter on 'information'
Lines 98/99 - 'Any participant could decline to take part in the survey after consenting if they so wished.' Is this what you meant?
Content:
Lines 100-105 - it would be better from the reader's point of view to see the whole questionnaire, possibly as an appendix? This list does not say precisely what was asked.
Figures 1-6 and 10-21 have not been labelled so they are not easy to follow. Please add labels throughout.
Table 2: It would be better if the description of what the table contains is moved so it appears just above the table. This keeps the flow better.
Discussion:
Line 159 - Why did you make this sweeping conclusion? Maybe the ones who did not complete the questionnaire decided that they couldn't be bothered? Or have you some other information that enables you to reach this conclusion?
Also, this only implies that 70% of your sample group use DFM - you cannot extrapolate this to the entire population.
Line 168 - '....using a different technique from that proposed by Cyriax'.
Line 169 - I don't know what you are trying to say in the sintence.
Line 170 'compared with'
Line 173/4 - 48 hours minimum or daily? Which question is this referring to?
Line 179 - 'given to'
Paragraph lines 183-189 - I don't understand what you are saying in this paragraph - perhaps it needs rewording?
Line 207 - too may commas in this sentence
Line 209 - remove the word 'serious' - not needed.
Line 212 - perhaps it would be better to rephrase this on the lines that the current thinking on treating tendinopathy is different.....etc.
Conclusion:
Line 226 - Previous chapter? Do you mean a previous section? If so, which one?
Your conclusion is based on a survey of 90 therapists. You can't apply your conclusion to the whole population based on your survey - you can only suggest that this might apply to the population.
Author Response
Firstly, we would like to thank you for spending time to read and comment to our paper. Your suggestions have been very helpful and had been taken in serious concern. In our try to improve our paper we have corrected all the language errors you have suggested and few more.
Introduction comments: the Cochrane review you point out was already included in the text (citation no 9)
DFM have been defined in main body
Physiotherapist have been replaced with physical therapist
Methods: Aim changed to aims and physiotherapist to physical therapist
Sentence rephrased: The questionnaire used closed-ended questions with an open free-text comments section in each question. (line 80)
About the sample selection: the sample was found as said in lines 85-87. There was no randomization.
Rephrased line 100-101 according to your suggestion.
Content: Concerning your comment about the large number of figured we decided to illustrate only the questions where our sample’s answer was in conflict with the guidelines. We also changed the format from pie chart to columns in an effort to present our results in a better way.
The questionnaire has been translated and added as an appendix.
Tables have been merged, redesigned and repositioned.
Discussion: This most probably suggests (line 127-128)
70% of our sample (line 129)
Changed have been done according to the suggestions.
Reviewer 3 Report
Reviewer comments
General comments
This is a good study using questionnaire surrey to investigate the application pattern of clinical Cypriot physiotherapists about DFM and their compliance to the guidelines suggested by Cyriax. 90 Cypriot physiotherapists were recruited. The questionnaire consisted of 18 multiple choice questions and a table with 6 sub-questions. The results showed that 70% of the physiotherapists declared that they perform DFM in their daily practice. They answered 11 of the 19 technical questions in compliance with the guidelines. The percentage compliance to Cyriax guidelines is 58% and 62.5% for chronic conditions. This study finding showed the evidence of neither the efficacy of TFM neither the damage of TFM. The conclusion might be little over-interpreted.
Forty percentage of non-compliance to Cyriax guidelines may point to the need of improvement for the guidelines through continuous education of professional body, which means that the technique proposed by Cyriax did not provide the clinicians with the adequate therapeutic results, underpinned by the hypothesis that each therapist would find the best efficacy of the treatment due to clinic’s management. If possible, further proposed suggestions would be helpful for this improvement, although prohibiting it is one of the choices and has insufficient evidence.
Specific comments
1. The structure of the content could be little modified. For example, the aim of the study in Materials and Methods may merge into the last paragraph of Introduction.
2. For more detail, please see the pdf file with the simple word-proofing.

Author Response
Firstly, we would like to thank you for spending time to read and comment to our paper. Your suggestions have been very helpful and had been taken in serious concern. In our try to improve our paper we have corrected some language errors. Regarding your comments, we have revised all the highlighted points. We did not merge the two sentences as you suggested. The reason is that the first one is referring to another study and the second one to ours. We hope that the changes we have done have strengthened the paper.
The figures have been reduced as other reviewers suggested. Now we only present the results of the questions where our sample did not comply to the guidelines.
Reviewer 4 Report
Dear authors,
Congratulations for this excellent work.
I think that the manuscript is well structured and presented. The objetives were clearly and well described. Methods section was correct. Results were clearly and concise. Discussion and conclusion were according with the results.
Author Response
Thank you for spending time to read and comment to our paper. Few changes have been done to the text according to the suggestions of the other 2 reviewers. Hopefully you will find the changed text as well structured and presented as the previous version.Round 2
Reviewer 1 Report
I do not think the manuscript has extensively been revised as suggested for scientific and academic rigour including completely rewriting the results. It is suggested that the authors refer to a similar study published in a good journal for English language, style etc.
I have highlighted the need for some of the edits in the Abstract below
Line 10-11: “widely used technique worldwide”- evidence for your claim?
Line 11: “in the matter of” – revisit English language grammar & style
Line 11: “according to Cyriax” – add a comma after the word Cyriax
Line 12: consider replacing the word “required” with “desired” – there’s no guarantee of achieving therapeutic results using DFM
Line 13-16: use numerals to distinguish objectives of the study
Line 14-15: empirical based application patterns – could be rephrased for clarity e.g., application techniques and patterns of DFM?
Line 15-16: “prevalence of DFM use” – consider rephrasing “point prevalence of the use of DFM by…”
Line 16: “in the clinical practice of Cyprus” - consider rephrasing “in the clinical practice of physical therapists in Cyprus”
Line 17: what is meant by “local” – a geographical location in Cyprus?
Line 20: technical questions in compliance with the guidelines – rephrase ‘technical questions in relation to the compliance with the guidelines’
Line 21: application pattern – unclear
Lines 23-25: We suggest that DFM should not be in the first line of musculoskeletal treatments, at the moment. It should not be used either in the management of chronic musculoskeletal pain – the conclusion doesn’t fall in line with the purpose of the study. Also need to revisit how the sentence is phrased.
Likewise, there is a need to pick up the quality of writing in the entire manuscript
Author Response
Dear reviewer 1,
The manuscript has extensively been revised by MDPI English editing team. We hope that the manuscript in its new for match your expectasions.
Reviewer 2 Report
Thank you for listing the corrections made in this paper. Unfortunately there are still some minor grammatical errors which I have identified in most cases, but it still needs another thorough proof reading.
My general comment is that this study has produced some interesting information on a very old technique. You have identified how current practice seems to differ from the original guidance but I'm not sure that the conclusions you make are based on the study findings. I think you could really analyse why techniques differ from the original guidance by considering things such as the training and education of therapists, how techniques have evolved with new research evidence etc. You have only touched on the notion that the therapeutic basis of Cyriax has been challenged, but there might be lots of other reasons why it is either not used, or used in a modified form or in conjunction with other techniques.
I have listed specific comments below according to line numbers:
INTRO:
Line 30 - take out the comma after the word 'Although'.
Line 35 - This should read 'Produce both...' (take out the word 'of').
Line 37 - remove the word 'wrong' and replace with something like 'incorrect'.
Line 38-42 - I don't understand this section. Are you saying that you propose the term TFM and then subdivisions of this term would be GTM and DTF?
Line 43 - Start with 'To date...' or 'At the time of writing...'.
Line 58 - Are you referring to the technique used in the Portuguese study?
Lines 55-64 - I find this final section muddled and unclear. I'm not sure exactly what point you are making.
RESULTS:
Figs 1-7 - I'm not clear whether all these columns represent areas of conflict with the techniques of Cyriax?
Fig 6 - Spelling error - 'Accessible'.
DISCUSSION:
Line 130-131 - Why does the fact that some therapists didn't answer beyond question 4 lead you to believe that they don't use this technique? Surely the answer to question 4 tells you this? Did any of the respondents who stopped at question 4 answer 'yes' to this question and if so, why do you think they might not have continued?
131-133 - You are making another conclusion which seems to be based on opinion, rather than your evidence.
138 - What exactly do you mean by 'empirical model-based patterns'?
139 - this should read '...in a different way from...'
142-155 - you need to refer the reader back to table 1 for reference, otherwise your comparisons aren't clear.
145-146 - I don't find this section clear. The therapists are meeting the guideline of 48 hours, so I'm not sure what you are saying?
156- 162 - This paragraph isn't clear. I'm unsure what is being said here.
Line 190 - you need to say what the limitations of the study are e.g. questionnaire design, compliance, lack of randomisation etc. Perhaps you might also consider the fact that you could have just targeted your questionnaire at therapists who use these techniques?
CONCLUSIONS:
194 - You are not assuming that they answered 19 questions, you are stating that there were 19 questions for them to answer.
198-199 - I don't understand how you reached this last conclusion from the data you collected. Your questionnaire was primarily to collect data from therapists using Cyriax techniques so I'm not sure exactly what you are alluding to in this last sentence.
FINAL COMMENTS:
I think you need to think about what data you collected and what this actually tells you about therapeutic practice, rather than making assumptions. There are lots of reasons why techniques may differ from the Cyriax guidelines - not all of them within the therapists' control (e.g. time available for treatment, local policies and procedures). Also, think about what this might add to the existing body of knowledge. Have we moved on from this old set of techniques? You have touched on the fact that new knowledge exists which challenges the Cyriax guidelines so why not expand a little on this? This would be clinically more useful for the reader. These are just some suggestion to make the paper more applicable to therapists reading it.
Author Response
Dear reviewer 2,
The manuscript has extensively been revised by MDPI English editing team. We hope that the manuscript in its new for match your expectations.
We have taken in serious concern all of your comments and added to the manuscript.
Round 3
Reviewer 1 Report
I thought the manuscript overally looks much improved, although there still is room for minor improvements - examples include (but not limited to)
WORDINESS - line 52-53: "create, share, and analyze a questionnaire aiming to" - this can deleted from the purpose statement. CONCISE / CLARITY - line 126: " physical therapists of all ages" - all ages sounds vague; in lines 135, 147 - it'd be nice to consistently write "physical therapist" for clarity. TAKE AWAY MESSAGE(S) / FAR-REACHING CONCLUSION - line 196-197: I agree with the author's statement there but I think it needs to be stated towards the end of discussion (just prior to the conclusion) and not in the conclusion itself. The authors could use this as a chance to discuss the evidence for the said antiquated therapy within the broader context of manual therapy.
Author Response
Dear reviewers,
All the suggested improvements have been made.
Once again, we would like to thank you for your suggestions and comments.